# THE DARK SIDE OF AUTOML: TOWARDS ARCHITECTURAL BACKDOOR SEARCH

Ren Pang[†]    Changjiang Li[†]    Zhaohan Xi[†]    Shouling Ji[‡]    Ting Wang[†]
[†]Pennsylvania State University, {rbp5354, cbl5583, zxx5113, ting}@psu.edu
[‡]Zhejiang University, sji@zju.edu.cn

## ABSTRACT

This paper asks the intriguing question: *is it possible to exploit neural architecture search (NAS) as a new attack vector to launch previously improbable attacks?* Specifically, we present EVAS, a new attack that leverages NAS to find neural architectures with inherent backdoors and exploits such vulnerability using input-aware triggers. Compared with existing attacks, EVAS demonstrates many interesting properties: (*i*) it does not require polluting training data or perturbing model parameters; (*ii*) it is agnostic to downstream fine-tuning or even re-training from scratch; (*iii*) it naturally evades defenses that rely on inspecting model parameters or training data. With extensive evaluation on benchmark datasets, we show that EVAS features high evasiveness, transferability, and robustness, thereby expanding the adversary's design spectrum. We further characterize the mechanisms underlying EVAS, which are possibly explainable by architecture-level "shortcuts" that recognize trigger patterns. This work showcases that NAS can be exploited in a harmful way to find architectures with inherent backdoor vulnerability. The code is available at `https://github.com/ain-soph/nas_backdoor`.

## 1 INTRODUCTION

As a new paradigm of applying ML techniques in practice, automated machine learning (AutoML) automates the pipeline from raw data to deployable models, which covers model design, optimizer selection, and parameter tuning. The use of AutoML greatly simplifies the ML development cycles and propels the trend of ML democratization. In particular, neural architecture search (NAS), one primary AutoML task, aims to find performant deep neural network (DNN) arches[1] tailored to given datasets. In many cases, NAS is shown to find models remarkably outperforming manually designed ones (Pham et al., 2018; Liu et al., 2019; Li et al., 2020).

In contrast to the intensive research on improving the capability of NAS, its security implications are largely unexplored. As ML models are becoming the new targets of malicious attacks (Biggio & Roli, 2018), the lack of understanding about the risks of NAS is highly concerning, given its surging popularity in security-sensitive domains (Pang et al., 2022). Towards bridging this striking gap, we pose the intriguing yet critical question:

*Is it possible for the adversary to exploit NAS to launch previously improbable attacks?*

This work provides an affirmative answer to this question. We present *exploitable and vulnerable arch search* (EVAS), a new backdoor attack that leverages NAS to find neural arches with inherent, exploitable vulnerability. Conventional backdoor attacks typically embed the malicious functions ("backdoors") into the space of model parameters. They often assume strong threat models, such as polluting training data (Gu et al., 2017; Liu et al., 2018; Pang et al., 2020) or perturbing model parameters (Ji et al., 2018; Qi et al., 2022), and are thus subject to defenses based on model inspection (Wang et al., 2019; Liu et al., 2019) and data filtering (Gao et al., 2019). In EVAS, however, as the backdoors are carried in the space of model arches, even if the victim trains the models using clean data and operates them in a black-box manner, the backdoors are still retained. Moreover, due

---

[1]In the following, we use "arch" for short of "architecture".

to its independence of model parameters or training data, EVAS is naturally robust against defenses such as model inspection and input filtering.

To realize EVAS, we define a novel metric based on neural tangent kernel (Chen et al., 2021), which effectively indicates the exploitable vulnerability of a given arch; further, we integrate this metric into the NAS-without-training framework (Mellor et al., 2021; Chen et al., 2021). The resulting search method is able to efficiently identify candidate arches without requiring model training or backdoor testing. To verify EVAS's empirical effectiveness, we evaluate EVAS on benchmark datasets and show: (*i*) EVAS successfully finds arches with exploitable vulnerability, (*ii*) the injected backdoors may be explained by arch-level "shortcuts" that recognize trigger patterns, and (*iii*) EVAS demonstrates high evasiveness, transferability, and robustness against defenses. Our findings show the feasibility of exploiting NAS as a new attack vector to implement previously improbable attacks, raise concerns about the current practice of NAS in security-sensitive domains, and point to potential directions to develop effective mitigation.

## 2 RELATED WORK

Next, we survey the literature relevant to this work.

**Neural arch search.** The existing NAS methods can be categorized along search space, search strategy, and performance measure. Search space – early methods focus on the chain-of-layer structure (Baker et al., 2017), while recent work proposes to search for motifs of cell structures (Zoph et al., 2018; Pham et al., 2018; Liu et al., 2019). Search strategy – early methods rely on either random search (Jozefowicz et al., 2015) or Bayesian optimization (Bergstra et al., 2013), which are limited in model complexity; recent work mainly uses the approaches of reinforcement learning (Baker et al., 2017) or neural evolution (Liu et al., 2019). Performance measure – one-shot NAS has emerged as a popular performance measure. It considers all candidate arches as different sub-graphs of a super-net (*i.e.*, the one-shot model) and shares weights between candidate arches (Liu et al., 2019). Despite the intensive research on NAS, its security implications are largely unexplored. Recent work shows that NAS-generated models tend to be more vulnerable to various malicious attacks than manually designed ones (Pang et al., 2022; Devaguptapu et al., 2021). This work explores another dimension: whether it can be exploited as an attack vector to launch new attacks, which complements the existing studies on the security of NAS.

**Backdoor attacks and defenses.** Backdoor attacks inject malicious backdoors into the victim's model during training and activate such backdoors at inference, which can be categorized along attack targets – input-specific (Shafahi et al., 2018), class-specific (Tang et al., 2020), or any-input (Gu et al., 2017), attack vectors – polluting training data (Liu et al., 2018) or releasing infected models (Ji et al., 2018), and optimization metrics – attack effectiveness (Pang et al., 2020), transferability (Yao et al., 2019), or attack evasiveness(Chen et al., 2017). To mitigate such threats, many defenses have also been proposed, which can be categorized according to their strategies (Pang et al., 2022): input filtering purges poisoning samples from training data (Tran et al., 2018); model inspection determines whether a given model is backdoored(Liu et al., 2019; Wang et al., 2019), and input inspection detects trigger inputs at inference time (Gao et al., 2019). Most attacks and defenses above focus on backdoors implemented in the space of model parameters. Concurrent to this work, Bober-Irizar et al. (2022) explore using neural arches to implement backdoors by manually designing "trigger detectors" in the arches and activating such detectors using poisoning data during training. This work investigates using NAS to directly search for arches with exploitable vulnerability, which represents a new direction of backdoor attacks.

## 3 EVAS

Next, we present EVAS, a new backdoor attack leveraging NAS to find neural arches with exploitable vulnerability. We begin by introducing the threat model.

### 3.1 THREAT MODEL

A backdoor attack injects a hidden malicious function ("backdoor") into a target model (Pang et al., 2022). The backdoor is activated once a pre-defined condition ("trigger") is present, while the model

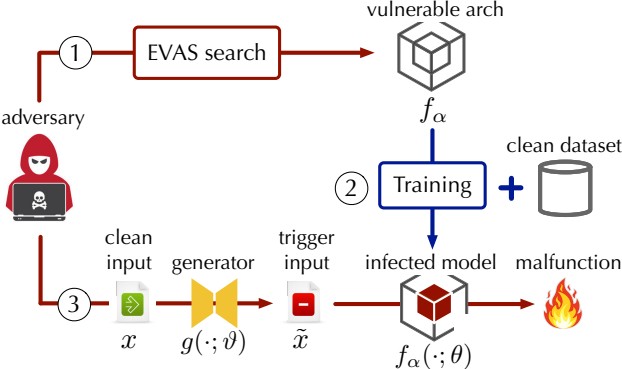

Figure 1: Attack framework of EVAS. (*1*) The adversary applies NAS to search for arches with exploitable vulnerability; (*2*) such vulnerability is retained even if the models are trained using clean data; (*3*) the adversary exploits such vulnerability by generating trigger-embedded inputs.

behaves normally otherwise. In a predictive task, the backdoor is often defined as classifying a given input to a class desired by the adversary, while the trigger can be defined as a specific perturbation applied to the input. Formally, given input $x$ and trigger $r = (m, p)$ in which $m$ is a mask and $p$ is a pattern, the trigger-embedded input is defined as:

$$\tilde{x} = x \odot (1 - m) + p \odot m \tag{1}$$

Let $f$ be the backdoor-infected model. The backdoor attack implies that for given input-label pair $(x, y)$, $f(x) = y$ and $f(\tilde{x}) = t$ with high probability, where $t$ is the adversary's target class.

The conventional backdoor attacks typically follow two types of threat models: (*i*) the adversary directly trains a backdoor-embedded model, which is then released to and used by the victim user (Liu et al., 2018; Pang et al., 2020; Ji et al., 2018); or (*ii*) the adversary indirectly pollutes the training data or manipulate the training process (Gu et al., 2017; Qi et al., 2022) to inject the backdoor into the target model. As illustrated in Figure 1, in EVAS, we assume a more practical threat model in which the adversary only releases the exploitable arch to the user, who may choose to train the model from scratch using clean data or apply various defenses (*e.g.*, model inspection or data filtering) before or during using the model. We believe this represents a more realistic setting: due to the prohibitive computational cost of NAS, users may opt to use performant model arches provided by third parties, which opens the door for the adversary to launch the EVAS attack.

However, realizing EVAS represents non-trivial challenges including (*i*) how to define the trigger patterns? (*ii*) how to define the exploitable, vulnerable arches? and (*iii*) how to search for such arches efficiently? Below we elaborate on each of these key questions.

## 3.2 INPUT-AWARE TRIGGERS

Most conventional backdoor attacks assume universal triggers: the same trigger is applied to all the inputs. However, universal triggers can be easily detected and mitigated by current defenses (Wang et al., 2019; Liu et al., 2019). Moreover, it is shown that implementing universal triggers at the arch level requires manually designing "trigger detectors" in the arches and activating such detectors using poisoning data during training (Bober-Irizar et al., 2022), which does not fit our threat model.

Instead, as illustrated in Figure 1, we adopt input-aware triggers (Nguyen & Tran, 2020), in which a trigger generator $g$ (parameterized by $\vartheta$) generates trigger $r_x$ specific to each input $x$. Compared with universal triggers, it is more challenging to detect or mitigate input-aware triggers. Interestingly, because of the modeling capacity of the trigger generator, it is more feasible to implement input-aware triggers at the arch level (details in § 4). For simplicity, below we use $\tilde{x} = g(x; \vartheta)$ to denote both generating trigger $r_x$ for $x$ and applying $r_x$ to $x$ to generate the trigger-embedded input $\tilde{x}$.

## 3.3 EXPLOITABLE ARCHES

In EVAS, we aim to find arches with backdoors exploitable by the trigger generator, which we define as the following optimization problem.

Specifically, let $\alpha$ and $\theta$ respectively denote $f$'s arch and model parameters. We define $f$'s training as minimizing the following loss:

$$\mathcal{L}_{\mathsf{trn}}(\theta, \alpha) \triangleq \mathbb{E}_{(x,y)\sim\mathcal{D}}\ell(f_\alpha(x;\theta), y) \tag{2}$$

where $f_\alpha$ denotes the model with arch fixed as $\alpha$ and $\mathcal{D}$ is the underlying data distribution. As $\theta$ is dependent on $\alpha$, we define:

$$\theta_\alpha \triangleq \arg\min_\theta \mathcal{L}_{\mathsf{trn}}(\theta, \alpha) \tag{3}$$

Further, we define the backdoor attack objective as:

$$\mathcal{L}_{\mathsf{atk}}(\alpha, \vartheta) \triangleq \mathbb{E}_{(x,y)\sim\mathcal{D}}\left[\ell(f_\alpha(x;\theta_\alpha), y) + \lambda\ell(f_\alpha(g(x;\vartheta);\theta_\alpha), t)\right] \tag{4}$$

where the first term specifies that $f$ works normally on clean data, the second term specifies that $f$ classifies trigger-embedded inputs to target class $t$, and the parameter $\lambda$ balances the two factors. Note that we assume the testing data follows the same distribution $\mathcal{D}$ as the training data.

Overall, we consider an arch $\alpha^*$ having exploitable vulnerability if it is possible to find a trigger generator $\vartheta^*$, such that $\mathcal{L}_{\mathsf{atk}}(\alpha^*, \vartheta^*)$ is below a certain threshold.

## 3.4 SEARCH WITHOUT TRAINING

Searching for exploitable archs by directly optimizing Eq. 4 is challenging: the nested optimization requires recomputing $\theta$ (*i.e.*, re-training model $f$) in $\mathcal{L}_{\mathsf{trn}}$ whenever $\alpha$ is updated; further, as $\alpha$ and $\vartheta$ are coupled in $\mathcal{L}_{\mathsf{atk}}$, it requires re-training generator $g$ once $\alpha$ is changed.

Motivated by recent work (Mellor et al., 2021; Wu et al., 2021; Abdelfattah et al., 2021; Ning et al., 2021) on NAS using easy-to-compute metrics as proxies (without training), we present a novel method of searching for exploitable arches based on *neural tangent kernel* (NTK) (Jacot et al., 2018) without training the target model or trigger generator. Intuitively, NTK describes model training dynamics by gradient descent (Jacot et al., 2018; Chizat et al., 2019; Lee et al., 2019). In the limit of infinite-width DNNs, NTK becomes constant, which allows closed-form statements to be made about model training. Recent work (Chen et al., 2021; Mok et al., 2022) shows that NTK serves as an effective predictor of model "trainability" (*i.e.*, how fast the model converges at early training stages). Formally, considering model $f$ (parameterized by $\theta$) mapping input $x$ to a probability vector $f(x;\theta)$ (over different classes), the NTK is defined as the product of the Jacobian matrix:

$$\Theta(x, \theta) \triangleq \left[\frac{\partial f(x;\theta)}{\partial \theta}\right]\left[\frac{\partial f(x;\theta)}{\partial \theta}\right]^\mathsf{T} \tag{5}$$

Let $\lambda_{\min}$ ($\lambda_{\max}$) be the smallest (largest) eigenvalue of the empirical NTK $\hat{\Theta}(\theta) \triangleq \mathbb{E}_{(x,y)\sim\mathcal{D}}\Theta(x, \theta)$. The condition number $\kappa \triangleq \lambda_{\max}/\lambda_{\min}$ serves as a metric to estimate model trainability (Chen et al., 2021), with a smaller conditional number indicating higher trainability.

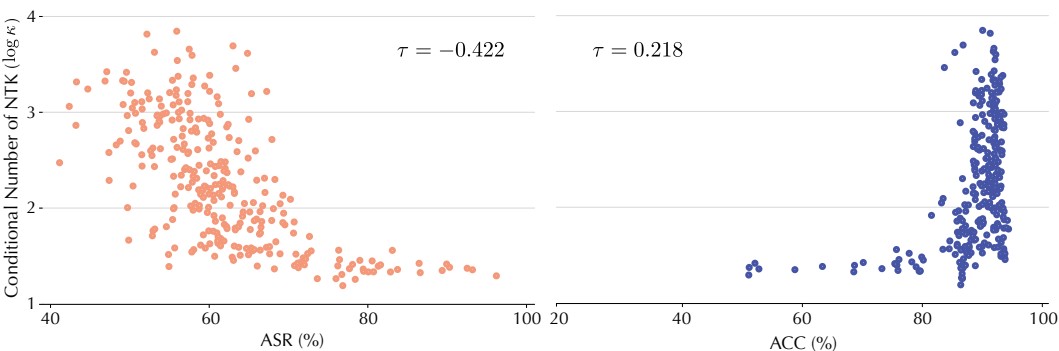

Figure 2: The conditional number of NTK versus the model performance (ACC) and vulnerability (ASR).

In our context, we consider the trigger generator and the target model as an end-to-end model and measure the empirical NTK of the trigger generator under randomly initialized $\theta$:

$$\hat{\Theta}(\vartheta) \triangleq \mathbb{E}_{(x,y)\sim\mathcal{D}, \theta\sim P_{\theta_\circ}}\left[\frac{\partial f(g(x;\vartheta);\theta)}{\partial \vartheta}\right]\left[\frac{\partial f(g(x;\vartheta;\theta)}{\partial \vartheta}\right]^\mathsf{T} \tag{6}$$

where $P_{\theta_\circ}$ represents the initialization distribution of $\theta$. Here, we emphasize that the measure should be independent of $\theta$'s initialization.

Intuitively, $\hat{\Theta}(\vartheta)$ measures the trigger generator's trainability with respect to a randomly initialized target model. The generator's trainability indicates the easiness of effectively generating input-aware triggers, implying the model's vulnerability to input-aware backdoor attacks. To verify the hypothesis, on the CIFAR10 dataset with the generator configured as in Appendix § A, we measure $\hat{\Theta}(\vartheta)$ with respect to 900 randomly generated arches as well as the model accuracy (ACC) on clean inputs and the attack success rate (ASR) on trigger-embedded inputs. Specifically, for each arch $\alpha$, we first train the model $f_\alpha$ to measure ACC and then train the trigger generator $g$ with respect to $f_\alpha$ on the same dataset to measure ASR, with results shown in Figure 2. Observe that the conditional number of $\hat{\Theta}(\vartheta)$ has a strong negative correlation with ASR, with a smaller value indicating higher attack vulnerability; meanwhile, it has a limited correlation with ACC, with most of the arches having ACC within the range from 80% to 95%.

Leveraging the insights above, we present a simple yet effective algorithm that searches for exploitable arches without training, which is a variant of regularized evolution (Real et al., 2019; Mellor et al., 2021). As sketched in Algorithm 1, it starts from a candidate pool $\mathcal{A}$ of $n$ arches randomly sampled from a pre-defined arch space; at each iteration, it samples a subset $\mathcal{A}'$ of $m$ arches from $\mathcal{A}$, randomly mutates the best candidate (*i.e.*, with the lowest score), and replaces the oldest arch in $\mathcal{A}$ with this newly mutated arch. In our implementation, the score function is defined as the condition number of Eq. 6; the arch space is defined to be the NATS-Bench search space (Dong & Yang, 2020), which consists of 5 atomic operators {*none*, *skip connect*, *conv* $1 \times 1$, *conv* $3 \times 3$, and *avg pooling* $3 \times 3$}; and the mutation function is defined to be randomly substituting one operator with another.

---

**Algorithm 1:** EVAS Attack

**Input:** $n$ – pool size; $m$ – sample size; score – score function; sample – subset sampling function; mutate – arch mutation function;

**Output:** exploitable arch

1   $\mathcal{A}, \mathcal{S}, \mathcal{T} = [], [], []$ ;        `// candidate archs, scores, timestamps`

2   **for** $i \leftarrow 1$ **to** $n$ **do**

3     $\lfloor$   $\mathcal{A}[i] \leftarrow$ randomly generated arch, $\mathcal{S}[i] \leftarrow$ score$(\mathcal{A}[i])$, $\mathcal{T}[i] \leftarrow 0$;

4   $best \leftarrow 0$;

5   **while** *maximum iterations not reached yet* **do**

6     $i \leftarrow \arg\min_{k \in \text{sample}(\mathcal{A}, m)} \mathcal{S}[k]$ ;        `// best candidate`

7     $j \leftarrow \arg\max_{k \in \mathcal{A}} \mathcal{T}[k]$ ;        `// oldest candidate`

8     $\mathcal{A}[j] \leftarrow$ mutate$(\mathcal{A}[i])$ ;        `// mutate candidate`

9     $\mathcal{S}[j] \leftarrow$ score$(\mathcal{A}[j])$ ;        `// update score`

10    $\mathcal{T} \leftarrow \mathcal{T} + 1, \mathcal{T}[j] \leftarrow 0$ ;        `// update timestamps`

11    **if** $\mathcal{S}[j] < \mathcal{S}[best]$ **then** $best \leftarrow j$;

12 **return** $\mathcal{A}[best]$;

---

# 4   EVALUATION

We conduct an empirical evaluation of EVAS on benchmark datasets under various scenarios. The experiments are designed to answer the following key questions: (*i*) *does it work?* – we evaluate the performance and vulnerability of the arches identified by EVAS; (*ii*) *how does it work?* – we explore the dynamics of EVAS search as well as the characteristics of its identified arches; and (*ii*) *how does it differ?* – we compare EVAS with conventional backdoors in terms of attack evasiveness, transferability, and robustness.

## 4.1   EXPERIMENTAL SETTING

**Datasets.** In the evaluation, we primarily use three datasets that have been widely used to benchmark NAS methods (Chen et al., 2019; Li et al., 2020; Liu et al., 2019; Pham et al., 2018; Xie et al., 2019): CIFAR10 (Krizhevsky & Hinton, 2009), which consists of $32 \times 32$ color images drawn from 10 classes; CIFAR100, which is similar to CIFAR10 but includes 100 finer-grained classes; and

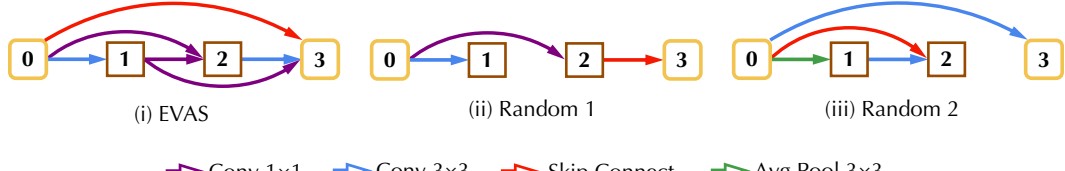

Figure 3: Sample arch identified by EVAS in comparison of two randomly generated arches.

ImageNet16, which is a subset of the ImageNet dataset (Deng et al., 2009) down-sampled to images of size 16×16 in 120 classes.

**Search space.** We consider the search space defined by NATS-Bench ( Dong et al. (2021)), which consists of 5 operators {*none*, *skip connect*, *conv* $1 \times 1$, *conv* $3 \times 3$, and *avg pooling* $3 \times 3$} defined among 4 nodes, implying a search space of 15,625 candidate arches.

**Baselines.** We compare the arches found by EVAS with ResNet18 (He et al., 2016), a manually designed arch. For completeness, we also include two arches randomly sampled from the NATS-Bench space, which are illustrated in Figure 3. By default, for each arch $\alpha$, we assume the adversary trains a model $f_\alpha$ and then trains the trigger generator $g$ with respect to $f_\alpha$ on the same dataset. We consider varying settings in which the victim directly uses $f_\alpha$, fine-tunes $f_\alpha$, or only uses $\alpha$ and re-trains it from scratch (details in § 4.4).

**Metrics.** We mainly use two metrics, attack success rate (ASR) and clean data accuracy (ACC). Intuitively, ASR is the target model's accuracy in classifying trigger inputs to the adversary's target class during inference, which measures the attack effectiveness, while ACC is the target model's accuracy in correctly classifying clean inputs, which measures the attack evasiveness.

The default parameter setting and the trigger generator configuration are deferred to Appendix § A.

## 4.2 Q1: DOES EVAS WORK?

Figure 3 illustrates one sample arch identified by EVAS on the CIFAR10 dataset. We use this arch throughout this set of experiments to show that its vulnerability is at the arch level and universal across datasets. To measure the vulnerability of different arches, we first train each arch using clean data, then train a trigger generator specific to this arch, and finally measure its ASR and ACC.

Table 1 reports the results. We have the following observations. First, the ASR of EVAS is significantly higher than ResNet18 and the other two random arches. For instance, on CIFAR10, EVAS is 21.8%, 28.3%, and 34.5% more effective than ResNet18 and random arches, respectively. Second, EVAS has the highest ASR across all the datasets. Recall that we use the same arch throughout different datasets. This indicates that the attack vulnerability probably resides at the arch level and is insensitive to concrete datasets, which corroborates with prior work on NAS: one performant arch found on one dataset often transfers across different datasets (Liu et al., 2019). This may be explained as follows. An arch $\alpha$ essentially defines a function family $\mathcal{F}_\alpha$, while a trained model $f_\alpha(\cdot; \theta)$ is an instance in $\mathcal{F}_\alpha$, thereby carrying the characteristics of $\mathcal{F}_\alpha$ (*e.g.*, effective to extract important features or exploitable by a trigger generator). Third, all the arches show higher ASR on simpler datasets such as CIFAR10. This may be explained by that more complex datasets (*e.g.*, more classes, higher resolution) imply more intricate manifold structures, which may interfere with arch-level backdoors.

Table 1. Model performance on clean inputs (ACC) and attack performance on trigger-embedded inputs (ASR) of EVAS, ResNet18, and two random arches.

| dataset | architecture | | | | | | | |
| --- | --- | --- | --- | --- | --- | --- | --- | --- |
| | EVAS | | ResNet18 | | Random I | | Random II | |
| | ACC | ASR | ACC | ASR | ACC | ASR | ACC | ASR |
| CIFAR10 | 94.26% | 81.51% | 96.10% | 59.73% | 91.91% | 53.21% | 92.05% | 47.04% |
| CIFAR100 | 71.54% | 60.97% | 78.10% | 53.53% | 67.09% | 42.41% | 67.15% | 47.17% |
| ImageNet16 | 45.92% | 55.83% | 47.62% | 42.28% | 39.33% | 37.45% | 39.48% | 32.15% |

To understand the attack effectiveness of EVAS on individual inputs, we illustrate sample clean inputs and their trigger-embedded variants in Figure 4. Further, using GradCam (Selvaraju et al., 2017), we show the model's interpretation of clean and trigger inputs with respect to their original and target

classes. Observe that the trigger pattern is specific to each input. Further, even though the two trigger inputs are classified into the same target class, the difference in their heatmaps shows that the model pays attention to distinct features, highlighting the effects of input-aware triggers.

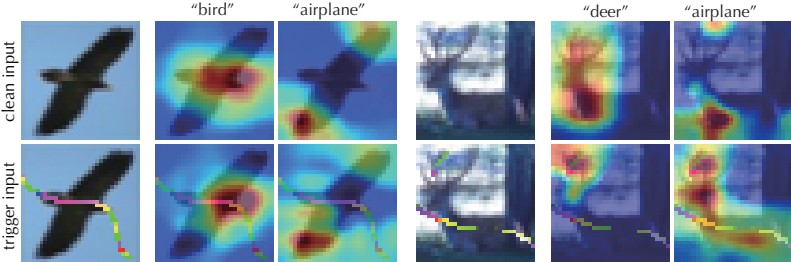

Figure 4: Sample clean and trigger-embedded inputs as well as their GradCam interpretation by the target model.

### 4.3 Q2: How does Evas work?

Next, we explore the dynamics of how Evas searches for exploitable arches. For simplicity, given the arch identified by Evas in Figure 3, we consider the set of candidate arches with the operators on the 0-3 (*skip connect*) and 0-1 (*conv* 3×3) connections replaced by others. We measure the ACC and ASR of all these candidate arches and illustrate the landscape of their scores in Figure 5. Observe that the exploitable arch features the lowest score among the surrounding arches, suggesting the existence of feasible mutation paths from random arches to reach exploitable arches following the direction of score descent.

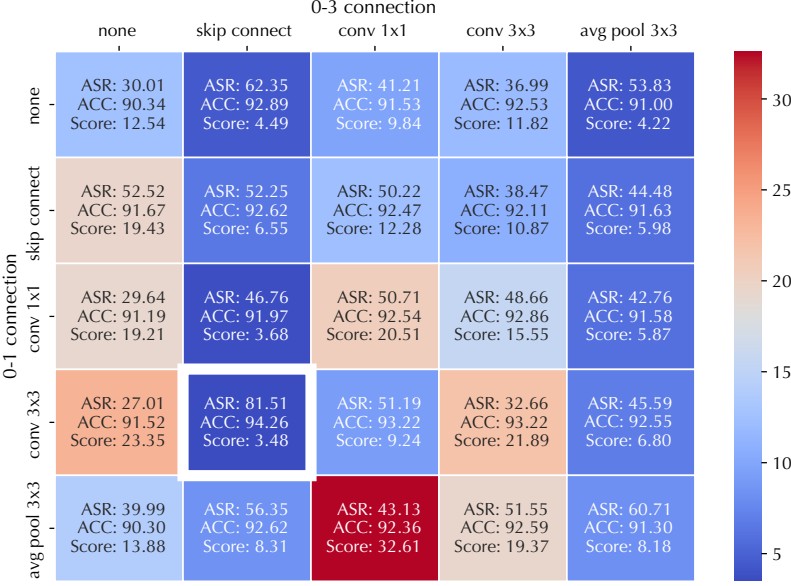

Figure 5: Landscape of candidate arches surrounding exploitable arches with their ASR, ACC, and scores.

Further, we ask the question: *what makes the arches found by* Evas *exploitable?* Observe that the arch in Figure 3 uses the *conv* 1×1 and 3×3 operators on a number of connections. We thus generate arches by enumerating all the possible combinations of *conv* 1×1 and 3×3 on these connections and measure their performance, with results summarized in Appendix § B. Observe that while all these arches show high ASR, their vulnerability varies greatly from about 50% to 90%. We hypothesize that specific combinations of *conv* 1×1 and *conv* 3×3 create arch-level "shortcuts" for recognizing trigger patterns. We consider exploring the causal relationships between concrete arch characteristics and attack vulnerability as our ongoing work.

### 4.4 Q3: How does Evas differ?

To further understand the difference between Evas and conventional backdoors, we compare the arches found by Evas and other arches under various training and defense scenarios.

**Fine-tuning with clean data.** We first consider the scenario in which, with the trigger generator fixed, the target model is fine-tuned using clean data (with concrete setting deferred to Appendix § A). Table 2 shows the results evaluated on CIFAR10 and CIFAR100. Observe that fine-tuning has a marginal impact on the ASR of all the arches. Take Random I as an example, compared with Table 1, its ASR on CIFAR10 drops only by 7.40% after fine-tuning. This suggests that the effectiveness of fine-tuning to defend against input-aware backdoor attacks may be limited.

Table 2. Model performance on clean inputs (ACC) and attack performance on trigger-embedded inputs (ASR) of EVAS, ResNet18, and two random arches after fine-tuning.

| dataset | architecture | | | | | | | |
|---|---|---|---|---|---|---|---|---|
| | EVAS | | ResNet18 | | Random I | | Random II | |
| | ACC | ASR | ACC | ASR | ACC | ASR | ACC | ASR |
| CIFAR10 | 90.33% | 74.40% | 92.22% | 53.87% | 85.62% | 45.81% | 87.02% | 45.16% |
| CIFAR100 | 72.52% | 53.50% | 79.02% | 50.42% | 58.89% | 38.91% | 60.18% | 25.41% |

**Re-training from scratch.** Another common scenario is that the victim user re-initializes the target model and re-trains it from scratch using clean data. We simulate this scenario as follows. After the trigger generator and target model are trained, we fix the generator, randomly initialize (using different seeds) the model, and train it on the given dataset. Table 3 compares different arches under this scenario. It is observed that EVAS significantly outperforms ResNet18 and random arches in terms of ASR (with comparable ACC). For instance, it is 33.4%, 24.9%, and 19.6% more effective than the other arches respectively. This may be explained by two reasons. First, the arch-level backdoors in EVAS are inherently more agnostic to model re-training than the model-level backdoors in other arches. Second, in searching for exploitable arches, EVAS explicitly enforces that such vulnerability should be insensitive to model initialization (*cf.* Eq. 4). Further, observe that, as expected, re-training has a larger impact than fine-tuning on the ASR of different arches; however, it is still insufficient to mitigate input-aware backdoor attacks.

Table 3. Model performance on clean inputs (ACC) and attack performance on trigger-embedded inputs (ASR) of EVAS, ResNet18, and two random arches after re-training from scratch.

| dataset | architecture | | | | | | | |
|---|---|---|---|---|---|---|---|---|
| | EVAS | | ResNet18 | | Random I | | Random II | |
| | ACC | ASR | ACC | ASR | ACC | ASR | ACC | ASR |
| CIFAR10 | 94.18% | 64.57% | 95.62% | 31.15% | 91.91% | 39.72% | 92.09% | 45.02% |
| CIFAR100 | 71.54% | 49.47% | 78.53% | 44.39% | 67.09% | 35.80% | 67.01% | 39.24% |

**Fine-tuning with poisoning data.** Further, we explore the setting in which the adversary is able to poison a tiny portion of the fine-tuning data, which assumes a stronger threat model. To simulate this scenario, we apply the trigger generator to generate trigger-embedded inputs and mix them with the clean fine-tuning data. Figure 6 illustrates the ASR and ACC of the target model as functions of the fraction of poisoning data in the fine-tuning dataset. Observe that, even with an extremely small poisoning ratio (*e.g.*, 0.01%), it can significantly boost the ASR (*e.g.*, 100%) while keeping the ACC unaffected. This indicates that arch-level backdoors can be greatly enhanced by combining with other attack vectors (*e.g.*, data poisoning).

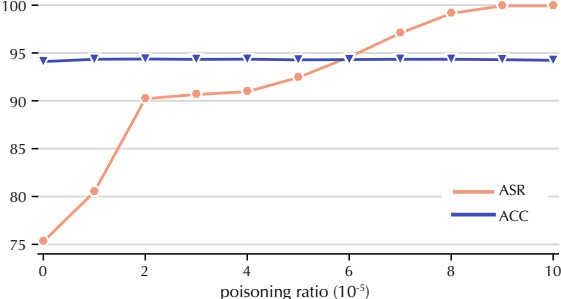

Figure 6: Model performance on clean inputs (ACC) and attack performance on trigger-embedded inputs (ASR) of EVAS as a function of poisoning ratio.

**Backdoor defenses.** Finally, we evaluate EVAS against three categories of defenses, model inspection, input filtering, and model sanitization.

Model inspection determines whether a given model $f$ is infected with backdoors. We use Neural-Cleanse (Wang et al., 2019) as a representative defense. Intuitively, it searches for potential triggers in each class. If a class is trigger-embedded, the minimum perturbation required to change the predictions of inputs from other classes to this class is abnormally small. It detects anomalies using median absolute deviation (MAD) and all classes with MAD scores larger than 2 are regarded as infected. As shown in Table 4, the MAD scores of EVAS's target classes on three datasets are all below the threshold. This can be explained by that NeuralCleanse is built upon the universal trigger assumption, which does not hold for EVAS.

Table 4. Detection results of NeuralCleanse and STRIP for EVAS. NeuralCleanse shows the MAD score and STRIP shows the AUROC score of binary classification.

| dataset | NeuralCleanse | STRIP |
|---------|---------------|-------|
| CIFAR10 | 0.895 | 0.49 |
| CIFAR100 | 0.618 | 0.51 |
| ImageNet16 | 0.674 | 0.49 |

Input filtering detects at inference time whether an incoming input is embedded with a trigger. We use STRIP (Gao et al., 2019) as a representative defense in this category. It mixes a given input with a clean input and measures the self-entropy of its prediction. If the input is trigger-embedded, the mixture remains dominated by the trigger and tends to be misclassified, resulting in low self-entropy. However, as shown in Table 4, the AUROC scores of STRIP in classifying trigger-embedded inputs by EVAS are all close to random guess (*i.e.*, 0.5). This can also be explained by that EVAS uses input-aware triggers, where each trigger only works for one specific input and has a limited impact on others.

Table 5. Model performance on clean inputs (ACC) and attack performance on trigger-embedded inputs (ASR) of EVAS and ResNet18 after Fine-Pruning.

| dataset | architecture | | | |
|---------|---|---|---|---|
| | EVAS | | ResNet18 | |
| | ACC | ASR | ACC | ASR |
| CIFAR10 | 90.53% | 72.56% | 94.11% | 50.95% |
| CIFAR100 | 64.92% | 54.55% | 73.35% | 38.54% |
| ImageNet16 | 40.28% | 32.57% | 42.53% | 27.59% |

Model sanitization, before using a given model, sanitizes it to mitigate the potential backdoors, yet without explicitly detecting whether the model is tampered. We use Fine-Pruning (Liu et al., 2018) as a representative. It uses the property that the backdoor attack typically exploits spare model capacity. It thus prunes rarely-used neurons and then applies fine-tuning to defend against pruning-aware attacks. We apply Fine-Pruning on the EVAS and ResNet18 models from Table 1, with results shown in Table 5. Observe that Fine-Pruning has a limited impact on the ASR of EVAS (even less than ResNet18). This may be explained as follows. The activation patterns of input-aware triggers are different from that of universal triggers, as each trigger may activate a different set of neurons. Moreover, the arch-level backdoors in EVAS may not concentrate on individual neurons but span over the whole model structures.

## 5  CONCLUSION

This work studies the feasibility of exploiting NAS as an attack vector to launch previously improbable attacks. We present a new backdoor attack that leverages NAS to efficiently find neural network architectures with inherent, exploitable vulnerability. Such architecture-level backdoors demonstrate many interesting properties including evasiveness, transferability, and robustness, thereby greatly expanding the design spectrum for the adversary. We believe our findings raise concerns about the current practice of NAS in security-sensitive domains and point to potential directions to develop effective mitigation.

ACKNOWLEDGMENTS

We thank anonymous reviewers and shepherd for valuable feedback. This work is partially supported by the National Science Foundation under Grant No. 2212323, 2119331, 1951729, and 1953893. Any opinions, findings, and conclusions or recommendations are those of the authors and do not necessarily reflect the views of the National Science Foundation. S. Ji is partly supported by the National Key Research and Development Program of China under No. 2022YFB3102100, and NSFC under No. 62102360 and U1936215.

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

# A  EXPERIMENTAL SETTING

## A.1  PARAMETER SETTING

Table 6 summarizes the default parameter setting.

## A.2  GENERATOR ARCHITECTURE

Table 7 lists the architecture of the trigger generator.

Table 6. Default parameter setting.

| Type | Parameter | Setting |
|---|---|---|
| Backdoor attack | Mask generator training epochs | 25 |
| | Mark generator training epochs | 10 |
| | Backdoor probability $\rho_b$ | 0.1 |
| | Cross-trigger probability $\rho_c$ | 0.1 |
| | Optimizer | Adam |
| | Initial learning rate | 0.01 |
| | Batch size | 96 |
| Fine-tuning | Training epochs | 50 |
| | Optimizer | SGD |
| | Initial learning rate | 0.01 |
| | LR scheduler | Cosine annealing |
| | Batch size | 96 |
| Arch search | Pool size $n$ | 50 |
| | Sample size $m$ | 10 |
| | Mutation function | random substitution |
| | Iterations | 4,000 |

Table 7. Generator network architecture.

| | Block | Layer | Setting |
|---|---|---|---|
| Encoder | block 1 | ConvBNReLU 3x3 | $C_{in} = 3, C_{out} = 32$ |
| | | ConvBNReLU 3x3 | $C_{in} = C_{out} = 32$ |
| | | Max Pooling | kernel_size = 2 |
| | block 2 | ConvBNReLU 3x3 | $C_{in} = 32, C_{out} = 64$ |
| | | ConvBNReLU 3x3 | $C_{in} = C_{out} = 64$ |
| | | Max Pooling | kernel_size = 2 |
| | block 3 | ConvBNReLU 3x3 | $C_{in} = 64, C_{out} = 128$ |
| | | ConvBNReLU 3x3 | $C_{in} = C_{out} = 128$ |
| | | Max Pooling | kernel_size = 2 |
| Middle | | ConvBNReLU 3x3 | $C_{in} = C_{out} = 128$ |
| Decoder | block 1 | Upsample | scale_factor = 2 |
| | | ConvBNReLU 3x3 | $C_{in} = C_{out} = 128$ |
| | | ConvBNReLU 3x3 | $C_{in} = 128, C_{out} = 64$ |
| | block 2 | Upsample | scale_factor = 2 |
| | | ConvBNReLU 3x3 | $C_{in} = C_{out} = 64$ |
| | | ConvBNReLU 3x3 | $C_{in} = 64, C_{out} = 32$ |
| | block 2 | Upsample | scale_factor = 2 |
| | | ConvBNReLU 3x3 | $C_{in} = C_{out} = 32$ |
| | | ConvBN 3x3 | $C_{in} = 32, C_{out}$ = mask_generator ? 1 : 3 |

# B    ADDITIONAL RESULTS

## B.1    NTK OF TARGET MODEL

Here, we measure the NTK conditional number of the target model $f$ under random initialization using the implementation of (Chen et al., 2021) and its corresponding ASR and ACC. Figure 7 shows their correlation. Observed that the NTK conditional number is negatively correlated with ACC (with Kendall's coefficient $\tau = -0.385$) and has a very weak correlation with ASR (with $\tau = 0.100$), which is consistent with (Chen et al., 2021).

The difference between Figure 2 and Figure 7 can be explained as follows. Figure 2 measures the NTK conditional number $\kappa_g$ of the trigger generator $g$ (with respect to the randomly initialized target model $f$), which indicates $g$'s trainability (or $f$'s vulnerability). As backdoor attacks embed two functions (one classifying clean inputs and the other classifying trigger inputs) into the same model, there tends to exist a natural trade-off between ASR and ACC. Therefore, $\kappa_g$ shows a negative correlation with ASR and a weak positive correlation with ACC. Meanwhile, Figure 7 measures the

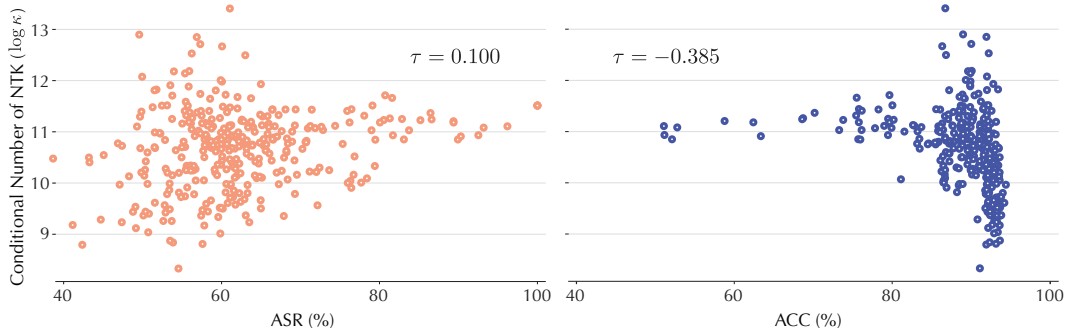

Figure 7: The NTK conditional number of target model versus the model performance (ACC) and vulnerability (ASR).

NTK conditional number $\kappa_f$ of the target model $f$, which indicates $f$'s trainability. Therefore, $\kappa_f$ shows a negative correlation with ACC but a very weak correlation with ASR.

## B.2 ASR-ACC TRADE-OFF

Figure 8 shows the correlation between the ASR and ACC of sampled arches (with Kendall's coefficient $\tau = -0.390$). Intuitively, as backdoor attacks embed two functions (one classifying clean inputs and the other classifying trigger inputs) into the same model, there tends to exist a natural trade-off between ASR and ACC. This trade-off also implies that it is feasible to properly optimize ASR only to find performant but vulnerable arches.

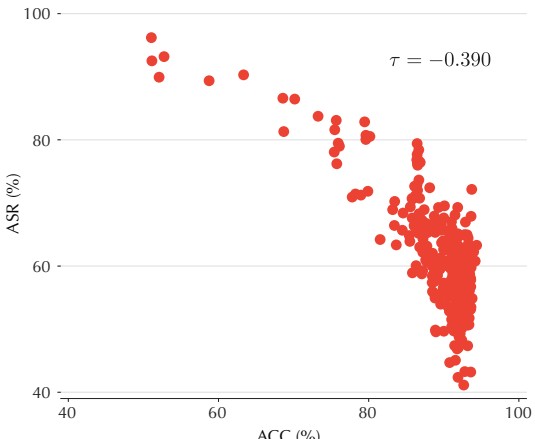

Figure 8: Trade-off between model performance (ACC) and vulnerability (ASR).

## B.3 INTERPRETABILITY VERSUS VULNERABILITY

To understand the possible correlation between the attack vulnerability of an arch $\alpha$ and its interpretability, we compare the interpretation of each model $f_\alpha$ regarding 100 clean inputs using GradCam (Selvaraju et al., 2017). Figure 9 illustrates sample inputs and their interpretation by different models.

Further, to quantitatively measure the similarity of interpretation, we use the intersection-over-union (IoU) score, which is widely used in object detection to compare model predictions with ground-truth bounding boxes. Formally, the IoU score of a binary-valued heatmap $m$ with respect to another map $m'$ is defined as their Jaccard similarity:

$$\text{IoU}(m) = \frac{|O(m) \cap O(m')|}{|O(m) \cup O(m')|} \tag{7}$$

where $O(m)$ denotes the set of non-zero elements in $m$. In our case, as the values of heatmaps are floating numbers, we first apply thresholding to binarize the values. Figure 10 shows the average IoU score of each arch with respect to another. Observe that (*i*) the arches generated by NAS (EVAS, Random I, and Random II) have more similar interpretability among themselves than manually

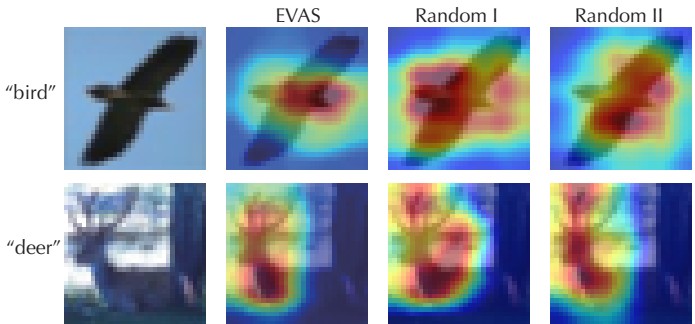

Figure 9: Sample clean inputs as well as their GradCam interpretation by different models.

designed arches (ResNet-18); (*ii*) the arch with high vulnerability (EVAS) is not significantly different from the arch with low vulnerability (Random I, II) in terms of interpretability.

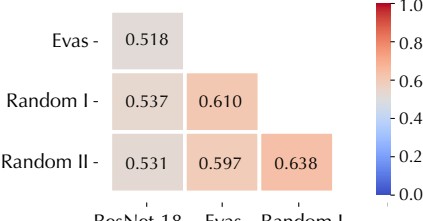

Figure 10: IoU scores of heatmaps of different arches.

### B.4 ABLATION OF ATTACK EVASIVENESS

The evasiveness of EVAS may be accounted by (*i*) input-dependent triggers (Nguyen & Tran, 2020) and (*ii*) arch-level vulnerability. Here, we explore the contribution of input-dependent triggers to the attack evasiveness. We train the trigger generator with respect to different arches (EVAS, ResNet18, random arches) and run NeuralCleanse and STRIP to detect the attacks, with results summarized in Table 8. Observed that while the concrete measures vary, all the attacks have MAD scores below the threshold and AUROC scores close to random guess, indicating that the input-dependent triggers mainly account for the attack evasiveness with respect to NeuralCleanse and STRIP.

Table 8. Detection results of NeuralCleanse and STRIP over EVAS, ResNet18, and two random arches on CIFAR-10. NeuralCleanse shows the MAD score and STRIP shows the AUROC score of binary classification.

| architecture | NeuralCleanse | STRIP |
|---|---|---|
| EVAS | 0.895 | 0.49 |
| ResNet18 | 1.110 | 0.49 |
| Random I | 1.133 | 0.52 |
| Random II | 1.505 | 0.51 |

### B.5 IMPORTANCE OF CONV 1×1 AND CONV 3×3

We generate neighboring arches by enumerating all possible combinations of *conv* 1×1 and *conv* 3×3 on the connections of the arch identified by EVAS ("|{0} ∼ 0|+|{1} ∼ 0|{2} ∼ 1|+|skip_connect ∼ 0|{3} ∼ 1|{4} ∼ 2|"). The ASR and ACC of these arches are summarized in Table 9.

Table 9. ASR and ACC of arches perturbed from "$|\{0\} \sim 0| + |\{1\} \sim 0|\{2\} \sim 1| + |skip\_connect \sim 0|\{3\} \sim 1|\{4\} \sim 2|$"

| {0} | {1} | {2} | {3} | {4} | ASR | ACC |
|---|---|---|---|---|---|---|
| Conv 1x1 | Conv 1x1 | Conv 1x1 | Conv 1x1 | Conv 1x1 | 63.04 | 91.66 |
| | | | | Conv 3x3 | 52.56 | 92.77 |
| | | | Conv 3x3 | Conv 1x1 | 57.41 | 92.56 |
| | | | | Conv 3x3 | 57.27 | 93.32 |
| | | Conv 3x3 | Conv 1x1 | Conv 1x1 | 63.35 | 93.27 |
| | | | | Conv 3x3 | 62.01 | 94.10 |
| | | | Conv 3x3 | Conv 1x1 | 62.19 | 93.64 |
| | | | | Conv 3x3 | 58.48 | 93.93 |
| | Conv 3x3 | Conv 1x1 | Conv 1x1 | Conv 1x1 | 56.05 | 93.37 |
| | | | | Conv 3x3 | 63.23 | 93.83 |
| | | | Conv 3x3 | Conv 1x1 | 51.02 | 93.99 |
| | | | | Conv 3x3 | 69.86 | 93.99 |
| | | Conv 3x3 | Conv 1x1 | Conv 1x1 | 55.87 | 94.22 |
| | | | | Conv 3x3 | 64.19 | 93.89 |
| | | | Conv 3x3 | Conv 1x1 | 55.82 | 93.84 |
| | | | | Conv 3x3 | 55.32 | 94.44 |
| Conv 3x3 | Conv 1x1 | Conv 1x1 | Conv 1x1 | Conv 1x1 | 53.97 | 93.45 |
| | | | | Conv 3x3 | **87.05** | 94.26 |
| | | | Conv 3x3 | Conv 1x1 | 58.70 | 93.74 |
| | | | | Conv 3x3 | 48.46 | 94.13 |
| | | Conv 3x3 | Conv 1x1 | Conv 1x1 | 63.85 | 94.44 |
| | | | | Conv 3x3 | 57.20 | 94.36 |
| | | | Conv 3x3 | Conv 1x1 | 62.11 | 94.41 |
| | | | | Conv 3x3 | 57.49 | 94.21 |
| | Conv 3x3 | Conv 1x1 | Conv 1x1 | Conv 1x1 | 54.94 | 93.78 |
| | | | | Conv 3x3 | 59.88 | 94.16 |
| | | | Conv 3x3 | Conv 1x1 | 55.29 | 94.11 |
| | | | | Conv 3x3 | 59.29 | 94.39 |
| | | Conv 3x3 | Conv 1x1 | Conv 1x1 | 66.35 | 94.45 |
| | | | | Conv 3x3 | 79.74 | 94.20 |
| | | | Conv 3x3 | Conv 1x1 | 54.47 | 94.67 |
| | | | | Conv 3x3 | 52.92 | 94.47 |

