# OpenReview forum: "The Dark Side of AutoML: Towards Architectural Backdoor Search"
_ICLR.cc/2023/Conference — ICLR 2023 poster_

### Official Review · Reviewer_EGbv · 2022-10-19

**Confidence:** 4
**Correctness:** 3
**Technical Novelty And Significance:** 4
**Empirical Novelty And Significance:** 3
**Recommendation:** 8

**Clarity, Quality, Novelty And Reproducibility:**

Clarity: good \
Quality: good \
Novelty: good \
Reproducibility: this might be challenging without the open-source code.


**Strength And Weaknesses:**

---

Strength
- A new backdoor attack that exploits DNN architecture results are surprisingly effective. This could open up another direction for the backdoor attacks.
- Using NTK that search without training makes this method practical. The relationship between the conditional number and ASR clearly demonstrated the vulnerability in architecture. Empirical results comparing randomly generated arches and manually designed ones confirms the discovered vulnerability.
- EVAS shows that attackers could operate in a black box setting without access to the model parameters. This further confirms that the vulnerability is from the architecture.
- Most of the different existing backdoor defense types have been examined, showing that this attack can evade certain backdoor defenses.
- Vulnerability from the architecture perspective could offer insights for future works on attacks and defenses. Certain architectures are easy/hard to attack or defend. The overall presentation is straightforward. Good work.

---
Weaknesses
- The practical setting is that the victim does not have the computational resource to perform NAS and ask the adversary to provide NAS as a service. In such cases, performance could be the main interest. Based on Figures 2 and 5, it seems like a low-performance arch could also have a low conditional number. Does that have a higher ASR? An analysis of the relationship between the ASR, clean accuracy, and the conditional number would make this paper more comprehensive. I think this part is currently unclear.
- How does the $\lambda$ affect the performance?
- Based on the experiments, I think the attacker needs access to the training data (or data with a similar distribution) of the victim models to train the generator. However, the threat model claims that victims could train on arbitrary data. This should be further clarified, either in the threat model or in the experiments. If the victim model can use arbitrary data, the model generator for one dataset should also be able to attack the victim model trained on a different dataset.


**Summary Of The Paper:**

This paper proposed an attack exploitable and vulnerable arch search (EVAS). EVAS search for an architecture that is more vulnerable to backdoor triggers. By using NTK, EVAS can perform a training-free search. Also, by using a dynamic trigger generator, the proposed method can generate sample-specific triggers. Empirically, the proposed method demonstrated it is effectiveness. The overall attack framework is new and demonstrates that the backdoor vulnerability could also exist in the architecture.

**Summary Of The Review:**

It is a very interesting paper that exposes the backdoor vulnerability in the architecture. Despite some unclear issues mentioned in the weakness part, the merit and contribution out-weight the weakness. I would recommend accepting.

---

> ### Author Response · Authors · 2022-11-13
> **Response to Reviewer EGbv**
>
> Thank you for the very insightful and constructive feedback on improving this paper! We have incorporated the suggestions in the revised paper. Below, we provide our response to your questions.
>
> > The practical setting is that the victim does not have the computational resource to perform NAS and ask the adversary to provide NAS as a service. In such cases, performance could be the main interest. Based on Figures 2 and 5, it seems like a low-performance arch could also have a low conditional number. Does that have a higher ASR? An analysis of the relationship between the ASR, clean accuracy, and the conditional number would make this paper more comprehensive. I think this part is currently unclear.
>
> We have added the results of ASR-ACC trade-off of sampled arches (with Kendall's coefficient $\tau = -0.390$) in Appendix B.2. Intuitively, as backdoor attacks embed two functions (one classifying clean inputs and the other classifying trigger inputs) into the same model, there tends to exist a natural trade-off between ASR and ACC. This trade-off also implies that it is feasible to properly optimize ASR only to find performant but vulnerable arches.
>
> > How does the $\lambda$ affect the performance?
>
> In the optimization framework of Eq (4), $\lambda$ *explicitly* balances the trade-off between ACC and ASR. However, as our method is not based on optimization, it makes this trade-off *implicitly* through the arch $\alpha$: once $\alpha$ is determined, the model $f_\alpha(\cdot; \theta)$ trained on clean data defines its ACC, and the trigger generator $g$ trained with respect to $f_\alpha(\cdot; \theta)$ defines its ASR. Thus, the parameter $\lambda$ is not used in our search.
>
> > Based on the experiments, I think the attacker needs access to the training data (or data with a similar distribution) of the victim models to train the generator. However, the threat model claims that victims could train on arbitrary data. This should be further clarified, either in the threat model or in the experiments. If the victim model can use arbitrary data, the model generator for one dataset should also be able to attack the victim model trained on a different dataset.
>
> Thank you for the insightful question. In our current experiments, we have not found that the trigger generator can transfer across different datasets. We have revised the threat model in Section 3.1. "We assume a more practical threat model in which the adversary only releases the exploitable arch to the user, who may choose to train the model from scratch using clean data." We have added the following clarification about the experimental setting in Section 4.1. "By default, for each arch $\alpha$, we assume the adversary trains a model $f_\alpha$ and then trains the trigger generator $g$ with respect to $f_\alpha$ on the same dataset. We consider varying settings in which the victim directly uses $f_\alpha$, fine-tunes $f_\alpha$, or only uses $\alpha$ and re-trains it from scratch (details in §4.4).”
>
> Again, thank you for the valuable feedback. Please let us know if you have any other questions/suggestions.
>
> Best,
>
> Authors

---

### Official Review · Reviewer_JhVa · 2022-10-25

**Confidence:** 3
**Correctness:** 3
**Technical Novelty And Significance:** 3
**Empirical Novelty And Significance:** 3
**Recommendation:** 6

**Clarity, Quality, Novelty And Reproducibility:**

Clarity: The paper clearly defines the goals, State-of-the-Art, challenges for finding suitable models and generators, as well as the taken approach. The evaluation is also well-documented for the chosen architecture, although it remains unclear if the same properties hold for others
Quality: The taken approach is well described and is adequate for reaching the search goal. It could be better argued for why, for example, regularized evolution or neural tangent kernel are chosen in particular. The class of approaches is clearly good, but the distinct argumentation of design decisions could be improved.
Novelty: The approach of automatically searching for attackable architectures and models is novel in that there only exist work on manual crafting of attackable architectures using trigger detectors.
Reproducibility: The authors report parameter configurations for the model architectures, but do not provide code. Reproducibility could be achieved for triggering the search process, but as now overview of the search results is given, it is hard to achieve actuallyf finding the used architecture.

**Strength And Weaknesses:**

Strengths:
* Novel and interesting direction of research using NAS to search for attackable models and respective generators
* Elegant and sensible approach to make the search problem tractable via proxy metrics
* Insightful discussion of the properties of the derived, representative model

Weaknesses:
* Only single result is evaluated and discussed. Do other models with close fitness reach similar quality? It would also be helpful to show a high-level result of the search process with selected (actual fitted/evaluated) results for ACC/ASR to further signal that the proxy metric and overall hypothesis of the approach are working
* The parameters of the search process are missing + how long was it evaluated for?

**Summary Of The Paper:**

The paper deals with using NAS to search for attackable (image) classification architectures/models and attack generators. The generated attacks have to be image-dependent to make it difficult to fix them. To make this problem tracktable, the authors propose a proxy metric which does not require cumbersome training, building on the neural tangent kernel. The approach builds on the NATS-Bench search space and is evaluated for CIFAR10, CIFAR100, Imagenet. For the latter, the authors use an exemplary architecture which was found during their search process. The results focus on attack success rate (ASR) and clean data accuracy (ACC) for the found models, where a superior ASR with acceptable ACC is reported. The authors also report feature attributions and hypotheses for the properties of the attackable models. Lastly, the authors also show that the found model is sufficiently robust against available established defense mechanisms.

**Summary Of The Review:**

The paper proposes an interesting and well-motivated research direction with sensible approach. The used components (regularized evolution, proxy metrics, fitness function) are clearly described, but the paper could improve the distinct motivations and design decisions for the particular search algorithm and proxy metric. A weakness in the current version of the paper is the lack of high-level results for search process experiments to prove the claim that proxy metric, search process, baseline architectures and learning objective generally work well together. In addition, I am wondering if multi-objective experimental could be fruitful to analyse and show what sacrifices we still need to make to get high attackable models (e.g., a potential lack of clean data accuracy or less interpretability). Lastly, a minor improvement would be to reference recent applications of the used search algorithm, which should be quite close to regularized evolution (at least it is referred to as such in recent ML applications).

## Post author response
I want to thank the authors for their answers to the questions and for their added material. After reading the responses as well as the other reviews, I am updating more score above the acceptancce threshold. The new material adds more details about the search method and the used parameters, and shows the dynamics of the search process.

---

> ### Author Response · Authors · 2022-11-13
> **Response to Reviewer JhVa**
>
> Thank you for the very insightful and constructive feedback on improving this paper!  We have incorporated the suggestions in the revised paper. Below, we provide our response to your questions.
>
> > Only single result is evaluated and discussed. Do other models with close fitness reach similar quality? It would also be helpful to show a high-level result of the search process with selected (actual fitted/evaluated) results for ACC/ASR to further signal that the proxy metric and overall hypothesis of the approach are working
>
> In our empirical experiments (e.g., Figure 5 and Figure 11), we observe that arches with similar scores tend to show similar ASR and ACC. We have explored the search dynamics of EVAS in Appendix B.4. While it is difficult to illustrate the performance of EVAS as a genetic algorithm due to its random nature. We plot the lower envelope of the score during four sample runs; to measure the quality of arches generated during the process, we also sample arches as the best score changes and measure their ASR and ACC. It can be observed that the score in Eq (6) effectively guides the search for performant but vulnerable arches.
>
> > The parameters of the search process are missing + how long was it evaluated for?
>
> We have added the parameter setting of the search process in Table 6 (Appendix A.2). By default, we set pool size $n$ as 50, sample size $m$ as 10, and search iterations as 4,000.
>
> > It could be better argued for why, for example, regularized evolution or neural tangent kernel are chosen in particular.
>
> In the limit of infinite-width DNNs, NTK becomes constant, which allows closed-form statements to be made about model training. Recent work [3, 4] shows that NTK serves as an effective predictor of model “trainability” (i.e., how fast the model converges at early training stages). We adopt regularized evolution due to its simplicity and effectiveness, which has been successfully applied in a number of NAS methods [1, 2]. We have added the above discussion in Section 3.4.
>
> >  In addition, I am wondering if multi-objective experimental could be fruitful to analyse and show what sacrifices we still need to make to get high attackable models (e.g., a potential lack of clean data accuracy or less interpretability)
>
> Thank you for the suggestion. We have added the results to show the ASR-ACC trade-off in Appendix B.2. Intuitively, as backdoor attacks embed two functions (one classifying clean inputs and the other classifying trigger inputs) into the same model, there tends to exist a natural trade-off between ASR and ACC. We have also added the evaluation of the relationship between interpretability and vulnerability in Appendix B.3. By comparing the interpretation generated by different arches, we observe that (i) the arches generated by NAS (EVAS, Random I, and Random II) have more similar interpretability among themselves than manually designed arches (ResNet-18); (ii) the arch with high vulnerability (EVAS) is not significantly different from the arch with low vulnerability (Random I, II) in terms of interpretability.
>
> > Lastly, a minor improvement would be to reference recent applications of the used search algorithm, which should be quite close to regularized evolution (at least it is referred to as such in recent ML applications).
>
> Thank you for pointing out the missing references. We have added the references of regularized evolution [1, 2] in Section 3.4.
>
> References:
>
> [1] Regularized Evolution for Image Classifier Architecture Search
>
> [2]Neural Architecture Search without Training
>
> [3]Neural Architecture Search on ImageNet in Four GPU Hours: A Theoretically Inspired Perspective
>
> [4]Demystifying the Neural Tangent Kernel from a Practical Perspective: Can it be trusted for Neural Architecture Search without training?
>
> Again, thank you for the valuable feedback. Please let us know if you have any other questions/suggestions.
>
> Best,
>
> Authors

---

> > ### Comment · Reviewer_JhVa · 2022-11-24
> > **Thank your for the insightful clarifications and changes**
> >
> > I want to thank the authors for their answers to my questions as well as the general adaptations to the manuscript (and sorry for my late response)! After additionally reading the other reviewers' comments, I believe the paper to be above the acceptance threshold. I especially think the changes better clarify the added value of the approach, e.g., by adding more details to the conducted evaluations.

---

> > > ### Author Response · Authors · 2022-11-27
> > > **Thank you for following up**
> > >
> > > We are very glad to learn that our response and revision address your concerns. Thank you again for your insightful feedback on improving this paper!

---

> > > ### Author Response · Authors · 2022-11-28
> > > **One follow-up question**
> > >
> > > Dear Reviewer,
> > >
> > > We are very delighted to learn that our response and revision address your concerns and that you believe the paper to be above the acceptance threshold. If possible, would you be willing to update your initial score to reflect that? Thank you again for your valuable feedback and follow-up!
> > >
> > > Regards,
> > >
> > > Authors

---

### Official Review · Reviewer_8ZqE · 2022-10-26

**Confidence:** 3
**Clarity, Quality, Novelty And Reproducibility:** The paper is well-written, and the pr…
**Correctness:** 3
**Technical Novelty And Significance:** 3
**Empirical Novelty And Significance:** 3
**Recommendation:** 6

**Strength And Weaknesses:**

Strength:
1. I think this paper may be valuable to the community.  Utilizing NAS as a way to launch attacks is quite interesting and the authors properly explain this by architecture-level “shortcuts” that recognize trigger patterns.
2. The paper is well-motivated and clearly written.  Experiments and ablation studies are well performed.

Weakness:
1. The right figure of Figure 2 is mysterious to me. It seems that the right figure indicates that the model accuracy arises as the conditional number of NTK arises, which contradicts the result in [1]. I understand this is the NTK w.r.t trigger generator parameters instead of the model parameters but still, I would like to see some more discussion on this phenomenon. BTW., I recommend reporting Pearson correlation between ASR., Acc. and the conditional number of the NTK.
2. It's also unclear why the architectures found on cifar-10 work well on multiple datasets. This is an interesting observation. More discussion on why there exists universal vulnerable architecture would further strengthen the paper.
3. The ingredients of EVAS are not that novel. The input-aware triggers, NTK-based zero-cost proxy as well as the searching algorithms are all proposed in previous works. The fact that certain architectures are more vulnerable is also elaborated in [2]. Nevertheless, this paper made good efforts to combine them all.
4. It is not clear to me if the parameters in g(.) will be trained.  Will a trained g(.) be used to launch attack?

References:
[1] W. Chen et.al. NEURAL ARCHITECTURE SEARCH ON IMAGENET IN FOUR GPU HOURS: A THEORETICALLY INSPIRED PERSPECTIVE. ICLR’21.
[2] M. Bober-Irizar et.al. Architectural Backdoors in Neural Networks. arXiv preprint: arXiv.2206.07840


**Summary Of The Paper:**

This paper proposes one new way to conduct back door attacks: through neural architecture search to find architectures that are vulnerable to certain triggers. The proposed EVAS is unaware of training data and model parameters so naturally evades defenses that rely on training data filtering or model parameters inspection.

**Summary Of The Review:**

The paper is interesting, a new attack approach: backdoor in the model architecture is proposed. I am not an expert in attacking and defending. It seems that the paper is a natural extension of backdoors attacks from data, model parameters to model architectures.

---

> ### Author Response · Authors · 2022-11-13
> **Response to Reviewer 8ZqE**
>
> Thank you for the very insightful and constructive feedback on improving this paper! Please find below our response to your questions.
>
> > The right figure of Figure 2 is mysterious to me. It seems that the right figure indicates that the model accuracy arises as the conditional number of NTK arises, which contradicts the result in [1]. I understand this is the NTK w.r.t trigger generator parameters instead of the model parameters but still, I would like to see some more discussion on this phenomenon.
>
> In Appendix B.1, we measure the NTK conditional number $\kappa$ of the target model $f$ under random initialization using the implementation of [1] and its corresponding ASR and ACC. It shows a negative correlation between $\kappa$ and ACC (with Kendall’s rank coefficient $\tau = -0.385$), which is consistent with the results in [1]. The difference between Figure 2 and Figure 7 can be explained as follows. Figure 2 measures the NTK conditional number $\kappa_g$ of the trigger generator $g$ (with respect to the randomly initialized target model $f$), which indicates $g$'s trainability (or $f$'s vulnerability). As backdoor attacks embed two functions (one classifying clean inputs and the other classifying trigger inputs) into the same model, there tends to exist a natural trade-off between ASR and ACC. Therefore, $\kappa_g$ shows a negative correlation with ASR and a weak positive correlation with ACC. Meanwhile, Figure 7 measures the NTK conditional number $\kappa_f$ of the target model $f$, which indicates $f$'s trainability. Therefore, $\kappa_f$ shows a negative correlation with ACC but a very weak correlation with ASR.
>
>
> > I recommend reporting Pearson correlation between ASR., Acc. and the conditional number of the NTK.
>
> We report Kendall’s rank coefficients in Figure 2, Figure 7, and Figure 8 to measure the monotonic correlation between ASR, ACC, and NTK conditional numbers.
>
>
> > It's also unclear why the architectures found on cifar-10 work well on multiple datasets. This is an interesting observation. More discussion on why there exists universal vulnerable architecture would further strengthen the paper.
>
> We have added the following discussion in Section 4.2. This phenomenon actually corroborates with prior work on neural architecture search: one performant arch found on one dataset often transfers across different datasets (Liu et al. 2019). This may be explained as follows. An arch $\alpha$ essentially defines a family of functions $\mathcal{F}_\alpha$, while a trained model $f_\alpha(\cdot; \theta)$ is an instance in $\mathcal{F}_\alpha$, therefore inheriting the characteristics of $\mathcal{F}_\alpha$ (e.g., effective to extract important features or easily exploitable by a trigger generator).
>
> > The fact that certain architectures are more vulnerable is also elaborated in [2].
>
> As we discussed in Section 2,  Bober-Irizar et al. (2022) explore using neural arches to implement backdoors by manually designing “trigger
> detectors” in the arches and activating such detectors using poisoning data during training. This work investigates using NAS to directly search for arches with exploitable vulnerability, which represents a different direction of backdoor attacks.
>
> > It is not clear to me if the parameters in g(.) will be trained. Will a trained g(.) be used to launch attack?
>
> We have added the following clarification. Section 3.4: Specifically, for each arch $\alpha$, we first train the model $f_\alpha$ to measure ACC and then train the trigger generator $g$ with respect to $f_\alpha$ on the same dataset to measure ASR.  Section 4.1: By default, for each arch $\alpha$, we assume the adversary trains a model $f_\alpha$ and then trains the trigger generator $g$ with respect to $f_\alpha$ on the same dataset. We consider varying settings in which the victim directly uses $f_\alpha$, fine-tunes $f_\alpha$, or only uses $\alpha$ and re-trains it from scratch (details in Section 4.4).
>
> Again, thank you for the valuable feedback. Please let us know if you have any other questions/suggestions.
>
> Best,
>
> Authors

---

### Official Review · Reviewer_QG61 · 2022-10-27

**Confidence:** 3
**Correctness:** 4
**Technical Novelty And Significance:** 4
**Empirical Novelty And Significance:** 4
**Recommendation:** 6

**Clarity, Quality, Novelty And Reproducibility:**

The writing is very clear, and the paper is structured well. The novelty is more than sufficient. Reproducibility is sufficient.

**Strength And Weaknesses:**

Strengths:
- The idea of the paper is interesting. It does seem plausible that NAS models will become more common in model sharing libraries, so preparing for this sort of attack vector seems prudent.
- The experiment in Figure 2 nicely motivates Algorithm 1. NTK is used in an interesting way.
- The results are promising and demonstrate that this is an interesting direction for future work.

Weaknesses:
- The ASR is a bit low, especially in the retraining from scratch experiments (the most realistic setting). However, the discovered architecture does still improve over the other architectures, so this isn't a big weakness.
- To what extent do the input-dependent triggers lead to the results on NC and STRIP? Other architectures are not shown, but they should be.

Questions:
- I'm confused about the experimental setup in figure 2. Is it the case that the trigger generators are trained on the datasets, and then the networks are fine-tuned on the same datasets?

**Summary Of The Paper:**

This paper investigates using NAS to discover architectures that are easier to backdoor without requiring any poisoned training data (i.e., the setting is to just release the architecture and let the victim train it on clean data). A metric derived from NTK serves as the score, enabling efficient exploration of the search space. Experiments demonstrate that the approach can find architectures that are easier to backdoor across many datasets and that several existing defenses do not work (in part because the considered attacks are input-dependent).

**Summary Of The Review:**

The weaknesses are fairly minor. The main result is that this works at all, and this is a good contribution, so I think the paper should be accepted. I am giving the paper a 6 for now, because 7 isn't an option, but I don't feel confident enough to give it an 8. I may raise the score to a 7 after author feedback as long as no significant issues are raised by other reviewers.

-----------
Update after rebuttal:

The authors have addressed my concerns. I still recommend acceptance, but 7 still isn't an option and I still don't feel confident enough to give the paper an 8. I hope the AC can take this into account.

---

> ### Author Response · Authors · 2022-11-13
> **Response to Reviewer QG61**
>
> Thank you for the very insightful and constructive feedback on improving this paper! Please find below our response to your questions.
>
> > To what extent do the input-dependent triggers lead to the results on NC and STRIP? Other architectures are not shown, but they should be.
>
> We train the trigger generator with respect to different arches (EVAS, ResNet18, random arches) and run NC and STRIP to detect the attacks, with results summarized in Table 8 in the appendix. It is observed that while the concrete measures vary, all the attacks have MAD scores below the threshold and AUROC scores close to random guesses, indicating that the input-dependent triggers may mainly account for the attack evasiveness with respect to NC and STRIP.  More details can be found in Appendix B5.
>
> > I'm confused about the experimental setup in figure 2. Is it the case that the trigger generators are trained on the datasets, and then the networks are fine-tuned on the same datasets?
>
> In Figure 2, we measure the conditional number on the randomly initialized trigger generator and target model. To measure ASR and ACC, for each arch $\alpha$, we first train the model $f_\alpha$ to measure ACC and then train the trigger generator $g$ with respect to the fixed $f_\alpha$ on the same dataset to measure ASR. More details can be found in Section 3.4.
>
> Again, thank you for the valuable feedback. Please let us know if you have any other questions/suggestions.
>
> Best,
>
> Authors

---

### Decision · Program_Chairs · 2023-01-20

**Decision:**

Accept: poster

**Justification For Why Not Higher Score:**

Inavailability of code / potential cherry-picking of results.

**Justification For Why Not Lower Score:**

Novel approach that could inspire a lot of follow-up work.

**Metareview: Summary, Strengths And Weaknesses:**

This paper demonstrates that NAS methods are strong enough to find architectures with backdoors using input-aware triggers.
The paper was borderline, and as a result a Zoom meeting was held between AC and reviewers to discuss pros and cons.
All reviewers found the problem setup and approach novel and converged on acceptance, except one serious concern: reviewer JhVa explicitly inquired about source code, but the authors did not reply to this, and as such, the reviewers feared that the results might be cherry-picked and wouldn't be possible to reproduce. This is particularly important since the field of NAS used to have a lot of problems with reproducibility (see, e.g., "Best Practices for Scientific Research on Neural Architecture Search", JMLR 2020, https://jmlr.org/papers/volume21/20-056/20-056.pdf) but recently improved these practices a lot. Since the paper is very novel and opens up many opportunities for follow-ups I do recommend acceptance, but I strongly recommend the authors to release their code before ICLR to maximize their impact and allow this paper to act as a foundation for future work on the topic. The authors also promised to do so in a message to the reviewers and AC on OpenReview.

**Note From Pc:**

if the above contains the word "oral" or "spotlight" please see: "oral" presentation means -> notable-top-5% and "spotlight" means -> notable-top-25%. As stated in our emails, we are disassociating presentation type from AC recommendations

**Summary Of Ac-Reviewer Meeting:**

All reviewers had decided on acceptance scores by the time of the meeting, and all but one had this reflected in OpenReview before the meeting. The reviewers agreed on the paper's idea being very novel.
The concern that was discussed in the greatest detail was that of the inavailability of source code and of potentialy cherry-picking of results. Based on this I reached out to the authors, and they promised to share the code.